# The Effects of Metakaolin on the Properties of Magnesium Sulphoaluminate Cement

**DOI:** 10.3390/ma17184567

**Published:** 2024-09-17

**Authors:** Lili Jiang, Zhuhui Li, Zhenguo Li, Dongye Wang

**Affiliations:** 1College of Civil Engineering and Architecture, Xiamen City University, Xiamen 361008, China; jianglili@xmcu.edu.cn (L.J.); wangdongye@xmcu.edu.cn (D.W.); 2College of Civil Engineering and Architecture, Harbin University of Science and Technology, Harbin 150080, China; 2321110029@stu.hrbust.edu.cn

**Keywords:** magnesium sulphoaluminate cement, metakaolin, compressive strength, water resistance, volumetric stability, microstructure

## Abstract

Magnesium sulphoaluminate (MSA) cement has good bonding properties and is suitable as an inorganic adhesive for repairing materials in civil engineering. However, there are still some problems with its use, such as its insufficient 1 day (d) strength and poor volumetric stability. This paper aims to investigate the influences of metakaolin (MK) on the physical and mechanical properties of magnesium sulphoaluminate (MSA) cement. The hydration products and microstructures of typical MSA cement samples were also analysed using X-ray diffraction (XRD), scanning electron microscopy (SEM), and energy-dispersive X-ray spectroscopy (EDS). The results showed that the addition of metakaolin reduces the fluidity and shortens the setting time of the MSA cement. The initial setting time and final setting time shortened maximally by 15–27 min and 25–48 min, respectively, with the addition of 10–30% metakaolin. Moreover, the compressive strength and flexural strength of the MSA cement improved significantly with the addition of 10–30% metakaolin at a curing age of 1 d. Compared with the compressive and flexural strengths of the control sample at 1 d, the compressive strengths of the modified samples showed obvious increases of 98%, 101%, and 109%, and the flexural strengths increased by 39%, 31%, and 26%, respectively, although they decreased slightly when the curing ages were 7 d, 14 d, and 28 d. The addition of 10% metakaolin improved the water resistance of the MSA cement immersed in water for 7 d and resulted in even higher water resistance at 28 d. The addition of 10–30% metakaolin improved the volumetric stability of the MSA cement with increasing dosages before 28 d of ageing. XRD and SEM-EDS analyses showed that the metakaolin accelerated the early hydration reaction and optimised the phase composition of the MSA cement. The results indicate that the addition of 10–20% metakaolin improved the strength after 1 d of ageing, water resistance, and volumetric stability of the MSA cement, providing theoretical support for the application of MAS cement as an inorganic bonding agent for repairing materials.

## 1. Introduction

Magnesium oxysulphate (MOS) cement has the advantages of excellent performance, energy savings, and emission reduction [1,2,3], and can mostly replace the usage of Portland cement. MOS cement is a ternary MgO-MgSO_4_-H_2_O cementitious system consisting of reactive MgO and a certain concentration of MgSO_4_ solution. At room temperature, the hydration of the MgO-MgSO_4_-H_2_O cementitious system of the MOS cement is insufficient, and the formation of the 3Mg(OH)_2_·MgSO_4_·8H_2_O phase in the hydration product is less than 50%, so the cement’s strength is poor [4,5].

In recent years, many scholars have conducted research into and made great progress for the MgO-MgSO_4_-H_2_O cementitious system of MOS cement. Wu et al. [6] found that the compressive strength and water resistance of MOS cement modified by phosphoric acid and phosphates (H_3_PO_4_, KH_2_PO_4_, K_3_PO_4_, or K_2_HPO_4_) improved significantly with the formation of a new needle-like hydration phase, 5Mg(OH)_2_·MgSO_4_·7H_2_O (517 phase). Runcevski et al. [7], at the Max Planck Institute for Solid-State Research, characterised and studied the structure of the 517 phase together with Yu and Wu. The same research group [8,9,10] also studied the modification effects of other additives (tartaric acid, sodium citrate, and citric acid) on MOS cements and the hydration mechanisms and compositions of the hydration products. The results showed that the incorporation of these additives changed the hydration process of the MgO and generated hydration products different from those of traditional MOS cement. The needle-like 517 phase is the main hydration product of the modified MOS cement, whereas flaked crystalline Mg(OH)_2_ is the main hydration product in traditional MOS cement. These changes in the hydration product category result in the property improvement of the modified MOS cement. Li et al. [11] comparatively investigated the effects of phosphoric acid and citric acid on MOS cement; the results indicated that the two additives yielded the same hydration products. However, the properties of the MOS cement modified by the citric acid were better than those of the phosphoric acid-modified cement because of its higher number of interlaced needle-shaped crystals and dense microstructure. Qin et al. [12] found that the properties of MOS cement modified by weak acids (citric acid, boric acid, and trisodium citrate) were enhanced significantly because of the formation of the 517 phase with needle-like whisker-shaped crystals in the hardened cement paste. Virginia et al. [13] investigated the effects of macromolecules on the hydration kinetics and microstructure of MOS cement. They found that macromolecules had a retardant effect on the hydration of the MgO and that the needle-shaped 517 phase was favoured at 20 °C for modifications of the MgO hydration kinetics. The research results of Zhou et al. [14] showed that the setting time of the MOS cement was prolonged, and the compressive strength and water resistance improved, significantly by the modification effect. Jin et al. [15] investigated the impacts of Cl^−^ on the properties of MOS cement. The results indicated that the crystallite size of the 517 phase was decreased, and that the content of the gel was increased by the influence of the Cl^−^. However, the addition of the proper content of Cl^−^ favoured improvements in the flexural strength and water resistance of the MOS cement.

According to their experimental exploration and production experience, Meng Chen et al. [16] added volcanic ash material to the MgO-MgSO_4_-H_2_O cementitious system of MOS cement, and under the coupling action of an alkali activator and a sulphate activator, magnesium sulphoaluminate (MSA) cement could be prepared. The raw materials of the MSA cement were lightly burnt magnesium oxide, magnesium sulphate solution, and bauxite, and the modifying agents were citric acid and aluminium sulphate. This reaction system is a new type of magnesium cement formed by MgO, as an alkaline excitation agent, and by magnesium sulphate and aluminium sulphate, as sulphate excitation agents. Different from MOS cement, the main hydration products of the MSA cement are magnesium sulphoaluminate hydrate (M-A-$\bar{S}$-H), magnesium silica hydrate (M-S-H), Al_2_(OH)_3_(AH_3_), and magnesium aluminosilicate hydrate (M-A-S-H). The physical and mechanical properties of this kind of magnesium cement are higher than those of the MOS cement, which provides a new way for conducting research on the modification of MOS cement. The excellent bonding properties of the MSA cement make it suitable as an inorganic bonding agent for repairing materials, but there are still problems, such as its insufficient strength within 1 d and large volumetric shrinkage. However, the admixture used in ordinary cement is not applicable in the MSA cement system and cannot effectively improve its early performance [17]. Therefore, it is necessary to carry out research to improve the early strength and reduce the shrinkage performance of the MSA cement.

Metakaolin is an anhydrous material that contains high amounts of aluminosilicates formed by the high-temperature dehydration of kaolin or calcined clay. It has a high volcanic ash activity and is often used as a concrete admixture [18]. The mineral composition and high activity of metakaolin are especially suitable for the hydration system of MSA cement. It is possible to accelerate the reaction by partially substituting the bauxite in MSA cement with metakaolin under the excitation of an alkali and a sulphate, thus further improving the early performance of the MSA cement, especially within 1 d. In this paper, the influences of metakaolin on the fluidity, setting time, compressive strength, flexural strength, water resistance, and volume stability of MSA cement were experimentally evaluated. On the other hand, typical MSA cement samples were selected for hydration products and microstructure measurements such as XRD and SEM-EDS analyses. This research can provide theoretical support for the application of MSA cement as an inorganic bonding agent for repairing materials.

## 2. Materials and Methods

### 2.1. Materials

#### 2.1.1. Magnesium Oxide

The magnesium oxide used in the present study was light-burnt MgO powder with a purity of 85% obtained from Haicheng, Liaoning Province, China. The content of active MgO used in this work was determined to be 61.6% by the standardized hydration method mentioned in Dong et al.’s report [19]. The chemical compositions of light -burnt MgO powder in wt.% are presented in Table 1.

The crystalline phases of the light-burnt MgO powder identified by XRD are shown in Figure 1. The XRD peaks of the crystalline MgO are clearly seen and small XRD peaks for MgCO_3_ appear in the pattern. The small amount of residual MgCO_3_ was due to the low calcination temperature of caustic magnesia.

The particle size distribution maps of the light-burnt MgO powder are shown in Figure 2; the average particle volume diameter of the magnesium oxide was 28.78 μm, D50 = 19.29 μm, and D90 = 72.95 μm.

#### 2.1.2. Bauxite and Metakaolin

In the present investigation, the bauxite was provided by the Henan Jiayuan Environmental Protection Materials Company, Henan Province, China. The key constituents were kaolinite and quartz, with the percentage of aluminium ranging from 40% to 70%, as presented in Table 2. The highly active metakaolin was supplied by the Tianzhijiao Kaolin Company, Nei Monggol Autonomous Region, China. The metakaolin has a kaolinite and quartz content exceeding 95%, displaying a thermodynamically mesostable state of high volcanic ash activity. The chemical compositions can be found in Table 2.

The XRD patterns of the bauxite and metakaolin are shown in Figure 3. The particle size distribution maps of the two raw materials are shown in Figure 3. It can be seen that the average particle volume diameter of the bauxite was 45.18 μm, D50 = 32.74 μm, D90 = 106.6 μm; and that of the metakaolin was 6.94 μm, D50 = 4.59 μm, D90 = 16.31 μm.

#### 2.1.3. Magnesium Oxysulfate and Modifier Additive

The magnesium oxysulfate (MgSO_4_·7H_2_O) employed was produced by an environmentally friendly building material manufacturer in Zhengzhou, Henan Province. The citric acid (C_6_H_8_O_7_·H_2_O) and aluminum sulfate (Al_2_(SO_4_)_3_) selected as modifier additives for the MSA cement were pure analytical reagent grade crystals obtained from Tianjin Biaozhunkeji Ltd., Tianjin, China.

### 2.2. Mix Ratio Design

The molar ratio of MgO:MgSO_4_:H_2_O in all the MAS cement mixtures were kept at 8:1:24, and the modifiers of citric acid and aluminum sulfate were 1% and 2% of the weight of the light-burned MgO, respectively. In the control sample (MSC) without metakaolin, the content of the bauxite was 45% of the weight of the light-burned MgO. Targeting the effects of metakaolin on the properties of MSA cement, the bauxite was partially substituted with 10–30% metakaolin, and the test amounts were 10% MK, 20% MK, and 30% MK, respectively.

For the production of the MSA cement pastes, the required amount of MgSO_4_·7H_2_O salt was dissolved in water to form a magnesium oxysulfate water solution at first. For the modified MSA, an appropriate amount of citric acid and aluminum sulfate were admixed with the MgSO_4_ solution to form a clear, uniform mixture. Then, the necessary amount of light-burned MgO, bauxite, and metakaolin powder were added into the MgSO_4_ solution and blended for a few minutes to produce the MSA cement paste. After the mixing operation, the pastes used for testing were poured into molds. The experimental design diagram is shown in Figure 4.

### 2.3. Testing Methods

The fluidity of the cement paste was tested according to the current Chinese National Standard GB/T 8077-2012 [20], and the test method of the cement setting time refers to the current Chinese National Standard GB/T 1346-2011 [21].

For each mixture, the cubic specimens with a size of 40 mm × 40 mm × 40 mm and 40 mm × 40 mm × 160 mm were cast into steel molds with vibration compaction. The strength development of the mixtures was recorded at 1, 3, 7, 14, and 28 days after air curing at a temperature of 20 ± 2 °C and under a relative humidity of 60 ± 5% in the curing room. The strength of each ratio and age was obtained from the average of three samples, referring to the current Chinese National Standard GB/T 17671-2021 [22].

The softening coefficient of cement is reliant on its ability to maintain strength after being submerged in water. The prepared test samples were placed in static water after 28 days of curing in a natural environment. Upon reaching the selected test age, the samples were dried to remove surface moisture and then tested for compressive strength and flexural strength. The softening coefficient was calculated according to Equation (1):(1)$$R_{n}=\frac{R(w,n)}{R(a,n)}$$
where *R_n_* refers to the softening coefficient, *R*(*w*,*n*) denotes the compressive or flexural strength after n days of soaking, and *R*(*a*,*n*) represents the compressive or flexural strength after n days of exposure to natural curing conditions.

The deformation of the MSA cement specimens with a size of 40 mm × 40 mm × 160 mm were studied by measuring the line ratio using a shrinkage dilatometer in a curing room maintained at a temperature of 20 ± 2 °C and a relative humidity of 60 ± 5%. The line ratio at *n* days (*ω*) was calculated according to the following equation:(2)$$\omega =\frac{L-L_{0}}{L}\times 100\%$$ where *ω* represents the shrinkage rate of the specimen; *L* represents the length of the specimen measured 24 h after curing; and *L*_0_ represents the length of the specimen at a specific age.

The crystalline phases were identified by the X-ray diffraction technique using Cu-Kα radiation. The morphology and microstructure of the MAS cement samples were characterized by scanning electron microscopy (SEM, Quanta 200 from Thermo Fisher Scientific (China) Co., Ltd., Shanghai, China) on the fractured surface with gold coating.

## 3. Results and Discussion

### 3.1. Effect of Metakaolin on the Fluidity and Setting Time of Magnesian Sulphoaluminate Cement

The influence of metakaolin on the fluidity and setting time of the MSA cement are demonstrated in Figure 5 and Figure 6. As illustrated in Figure 5, the fluidity of the cement paste decreased with increasing dosages of metakaolin. The fluidity measured at 70 mm when containing 10% metakaolin, which is 6.6% lower than that of the control sample. With the addition of 20% metakaolin, the fluidity was 67 mm, which is 10.6% lower than that of the control sample. The fluidity virtually ceased when the slurry contained 30% metakaolin, which is 16% lower than that of the control sample. In Figure 6, the graph demonstrates that the setting time of the MSA cement shortened with increases in the metakaolin content. The initial setting time and final setting time were shortened by a maximum 15–27 min and 25–48 min, respectively, with the addition of 10–30% metakaolin.

The metakaolin underwent high-temperature calcination, resulting in particles with a disordered structure and a larger specific surface area. The water requirement of the cement increases with the addition of metakaolin. Thus, the irregular particle morphology of metakaolin increases the viscosity and significantly reduces the fluidity of the mixture, as noted in reference [23]. The setting time of the MSA cement is mainly influenced by several factors, including the level and content of reactive magnesium oxide, the concentration of the magnesium sulfate solution, and the properties of metakaolin. When the level and content of reactive magnesium oxide and the concentration of the magnesium sulfate solution are constant, the properties of metakaolin become the primary factors affecting the setting time of the MSA cement. The setting time of the MSA cement decreases with increases in the amount of metakaolin, primarily due to the higher water demand ratio and high pozzolanic activity of metakaolin under the action of alkalis and sulfates, which accelerates the hydration reaction of the system. These findings are consistent with the results that the addition of metakaolin alone and the compounded addition of fly ash and metakaolin can shorten the setting time of magnesium oxychloride cement [24].

### 3.2. Effect of Metakaolin on the Strength of Magnesian Sulphoaluminate Cement

The compressive strength and flexural strength of the MSA cement with the addition of 10–30% metakaolin at 1, 3, 7, 14, and 28 days (d) are illustrated in Figure 7 and Figure 8.

From Figure 7, it can be seen that the compressive strength of the MSA cement was significantly enhanced by the addition of 10–30% metakaolin at the curing ages of 1 d and 3 d, and it increased with increases in the mixing dosage. Specifically, the compressive strength at 1 d showed an obvious increasement of 98%, 101%, and 109% compared with the control sample, respectively. However, it experienced a slight decrease when the curing age was 7 d, 14 d, and 28 d. In Figure 8, the flexural strength of the MSA cement at 1 d curing age improved obviously with the addition of 10–30% metakaolin, showing an increase of 39%, 31%, and 26% respectively, when compared with the control sample. At 3 d, 7 d, 14 d, and 28 d, the flexural strength decreased slightly with increases in the mixing metakaolin dosage. The results indicate that the compressive strength and flexural strength of the MSA cement at 1 d was evidently enhanced with the addition of 10–30% metakaolin.

There are two main reasons for why metakaolin is able to improve the early strength of MSA cement. On the one hand, the finer particles of metakaolin can effectively act as a filling agent in cement paste, thereby improving the pore structure [25]. On the other hand, metakaolin is an amorphous aluminosilicate mixture with high pozzolanic activity, and under the activation of alkalis and sulfates it can hydrate earlier and release hydration heat, which promotes the reaction rate of the system [26,27]. In the XRD analysis, the hydration reaction created a number of gel substances quickly at 1 d, including magnesium aluminosilicate hydrate (M-A-S-H) and magnesium silica hydrate (M-S-H), due to the action of the alkalis and sulfates. This process significantly improves the compactness of the system [28,29]. Thus, the strength at 1 d showed a noteworthy enhancement in comparison with the control sample. The insufficient hydration heat, along with the lack of alkalis and sulfates, resulted in a slight decrease in the compressive strength and flexural strength of the MSA cement at the curing age of 7 d, 14 d, and 28 d.

### 3.3. Effect of Metakaolin on the Water Resistance of Magnesian Sulphoaluminate Cement

The influence of metakaolin on the water resistance of MSA cement is demonstrated in Figure 9. The results show that the softening coefficient of the MSA cement with 10% metakaolin was higher than that of the control sample after soaking in water for 7 d, but it was slightly lower than the control sample’s softening coefficient at the soaking age of 28 d. The softening coefficient of mixtures with 20–30% metakaolin showed a lower value than that of the control sample at any soaking age. It is worth noting that the softening coefficients of mixtures with 10% and 20% metakaolin were all higher than 0.85 at the 28 d soaking period. These results indicate that the addition of 10% metakaolin improves the water resistance of the MSA cement when immersed in water for 7 d and it exhibits even higher water resistance at 28 d. Nevertheless, if the dose of metakaolin exceeds 10%, it leads to a decrease in the softening coefficient.

The improved water resistance of the MAS cement with the addition of 10% metakaolin after soaking in water for 7 days is attributed to the combined effects of the physical and chemical actions of metakaolin. The physical action is the filling effect of the fine particles of metakaolin in the pores, which decreases the total porosity and refines the pore structure [30]. The chemical action is the formation of a large amount of gel phase during the hydration of the metakaolin, which acts as a filling and wrapping agent. These two actions make it difficult for moisture to enter the interior of the cement, thereby weakening the bonding force between the hydration products. With increases in the metakaolin content, the cement slurry becomes more viscous, which affects the compaction of the hardened cement, and more larger-sized pores form in the interior, leading to a reduction in the cement water resistance [24,31].

### 3.4. Effect of Metakaolin on the Volume Stability of Magnesian Sulphoaluminate Cement

The influence of metakaolin on the shrinkage of MSA cement is depicted in Figure 10. As depicted in the diagram, the addition of metakaolin markedly enhances the volume stability of the MSA cement. At a curing age of 1 day, the shrinkage reductions of the MSA cement with 10–30% metakaolin were 15.1%, 5.6%, and 4.2% respectively, compared to the control sample. The MSA cement with 10–30% metakaolin represented a reduction in shrinkage of 20.3%, 22.7%, and 36.1% at 28 d compared with the sample without metakaolin. On the one hand, the reduction in cement shrinkage is due to the gel products filling the internal pores, which optimizes the microstructure of the cement stone [12]. On the other hand, the contraction is compensated for by the certain expansion of the hydration products [32]. The mechanism of shrinkage reduction in MSA cement with metakaolin needs to be further studied.

### 3.5. XRD Analysis

The XRD patterns of the MSA cement and MSA cement with 30% metakaolin at 1 d and 28 d are shown in Figure 11. It can be seen from the figure that the main hydration products of the MSA cement are the magnesium sulphoaluminate hydrate (M-A-$\bar{S}$-H) crystalline phase, as well as the flat-shape-like product magnesium silica hydrate (M-S-H), Al_2_(OH)_3_(AH_3_), and M-A-S-H products with poor crystallinity. These hydration products differ from those of MOS cement [33,34,35]. In addition, the system contains a small amount of Mg(OH)_2_, residual MgO, and MgCO_3_ impurities, and peaks corresponding to quartz from the bauxite. No new hydration products were generated after the addition of metakaolin. Figure 11 demonstrates that the M-A-$\bar{S}$-H crystal diffraction peaks exhibited similar peak intensities at the characteristic angles of 15.78°, 17.21°, and 18.86° for both groups after 1 day of curing. However, the intensity of the characteristic peaks corresponding to the flat-shape-like products in the MSA cement with 30% metakaolin was higher than that of the control sample.

### 3.6. SEM-EDS Analysis

The morphologies of MSA cement and MSA cement with 30% metakaolin at 1 d and 28 d are shown in Figure 12. It can be seen from the figures that in the control sample, after 1 day, a substantial quantity of M-A-$\bar{S}$-H crystals and a few M-S-H gel products were generated. Additionally, the needle-shape crystals depicted in Figure 12a exhibit significantly more bulk and more interlocking with each other between crystals. After 28 days, the system mixed with 30% metakaolin produced numerous disordered gel products. The M-A-$\bar{S}$-H crystals were generated with flocculent gels filling the tops of the crystals and the inter-pore spaces. In the control sample, the M-A-$\bar{S}$-H crystals grew uniformly and were longer in length. Many small clumps of flocculent gel and some sheets of gel formed around them were M-S-H and M-A-S-H phases. The addition of metakaolin accelerated the hydration rate of the MSA cement, resulting in the formation of longer interwoven needle-like crystals and gels during the early stage compared to the control sample.

As illustrated by Figure 13 and Table 3, the needle-shaped crystals observed in the system at 28 days belonged to the M-A-$\bar{S}$-H category, primarily composed of Al, S, O, and Mg elements. There was a large amount of sheet material consisting of Mg and O as Mg(OH)_2_ in the system. The gel substance in Figure 13 consists mainly of Al, Si, O, and Mg, from which it can be determined to be M-A-S-H.

## 4. Conclusions

This study focuses on the effects of metakaolin on the fluidity, setting time, compressive strength, flexural strength, water resistance, and volume stability of MSA cement. The following results can be drawn:(1)The addition of metakaolin reduces the fluidity and shortens the setting time of MSA cement. With the addition of 10–30% metakaolin, the fluidity was reduced by 6.6–16% compared with the control sample due to the irregular morphology of the grains. The initial setting time and final setting time shortened by 15–27 min and 25–48 min, respectively, for the metakaolin additions in the hydration reaction earlier.(2)The early strength of the MSA cement is significantly improved by the suitable addition of metakaolin. At a 1 day curing age of MSA cement with a dosage of 10–30% metakaolin, the compressive strength and flexural strength were enhanced by 98% and 39%, respectively, because metakaolin accelerates the early hydration reaction and generates more gel-like hydration products. Nevertheless, the strength decreased slightly when the curing age was 7 d, 14 d, and 28 d.(3)The water resistance of the MSA cement was improved by the addition of 10–20% metakaolin within the first 3 d and decreased slightly at other curing ages. However, the softening coefficients of the mixtures with 10% and 20% metakaolin were all higher than 0.85 before the 28 d soaking period. The incorporation of 30% metakaolin reduced water resistance at all ages. The addition of 10–30% metakaolin caused an improvement on the volume stability of the MSA cement with increasing dosages before 28 d of age. At a curing age of 1 day, the shrinkage reductions in the MSA cement with 10–30% metakaolin were 15.1%, 5.6% and 4.2% compared to the control sample, which may be due to the more rational phase composition and microstructure. The mechanism of shrinkage reduction of the MSA cement with metakaolin needs to be further studied.(4)The results show that addition of 10% metakaolin could shorten the setting time and significantly improve the compressive strength, flexural strength, water resistance, and volume stability of MSA cement at1 d of age, while also maintaining good long-term performance. This finding may promote the application of MSA cement as an inorganic bonding agent for repairing materials.

## Figures and Tables

**Figure 1 materials-17-04567-f001:**
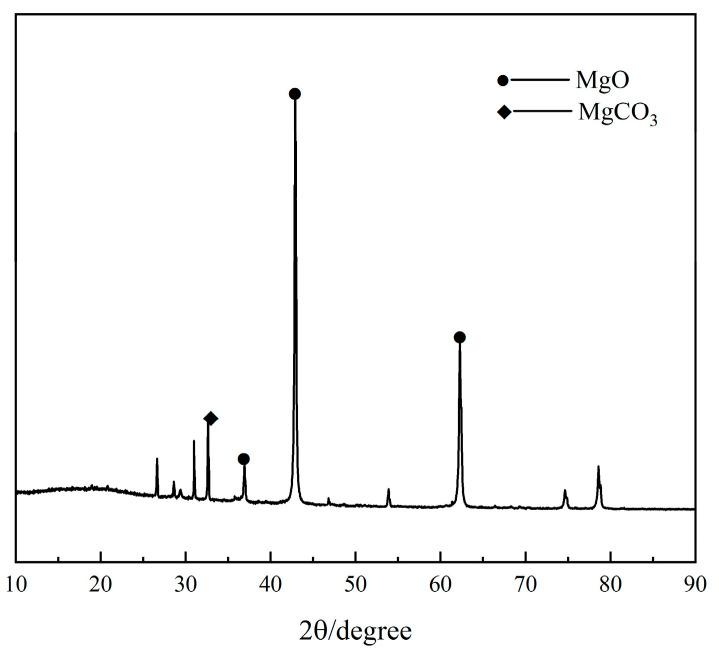
XRD pattern of magnesium oxide.

**Figure 2 materials-17-04567-f002:**
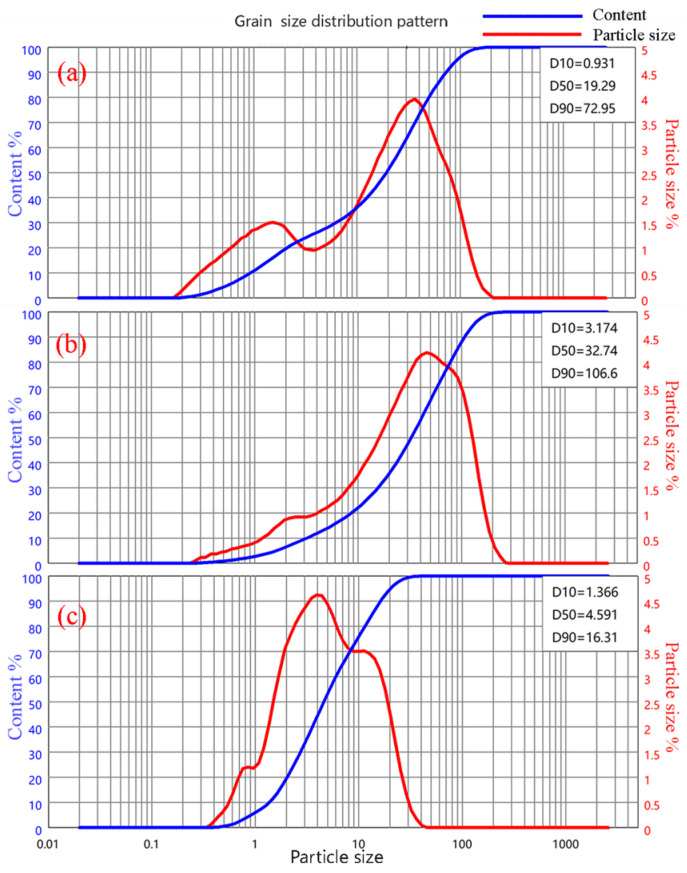
Laser particle size analysis curve of the raw materials: (**a**)oxide; (**b**) bauxite; (**c**) metakaolin.

**Figure 3 materials-17-04567-f003:**
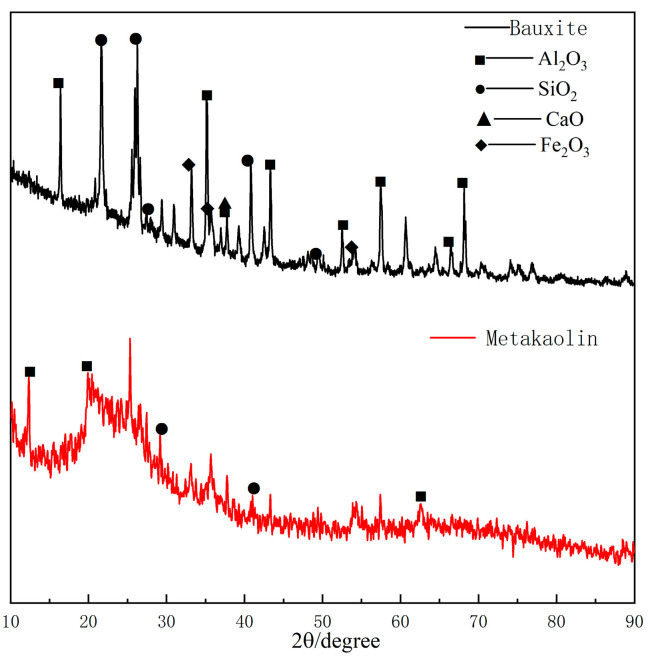
XRD pattern of the bauxite and Metakaolin.

**Figure 4 materials-17-04567-f004:**
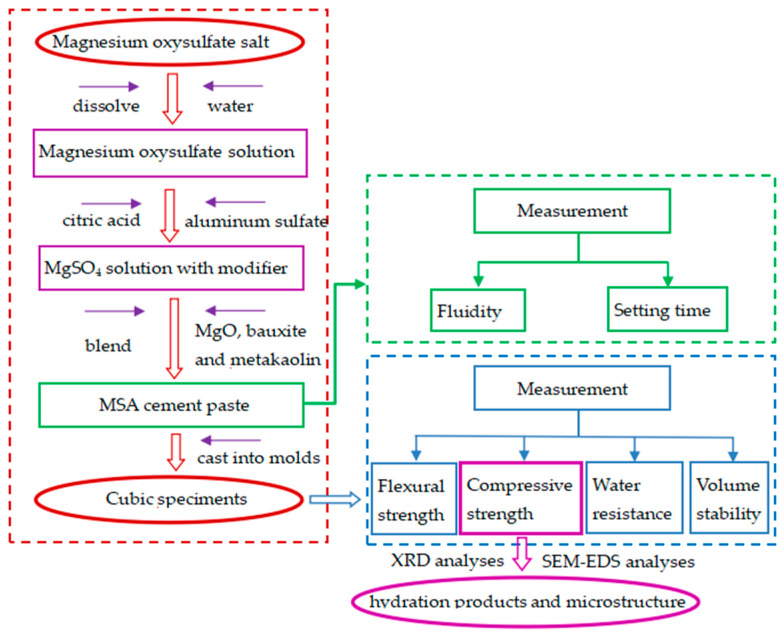
The experimental design diagram of the MSA cement.

**Figure 5 materials-17-04567-f005:**
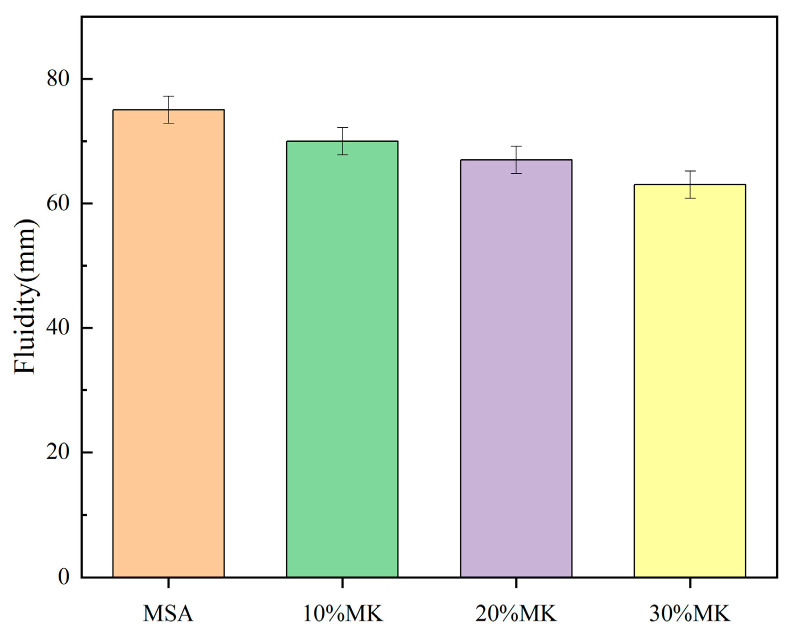
Effect of MK on fluidity of MSA cement slurry.

**Figure 6 materials-17-04567-f006:**
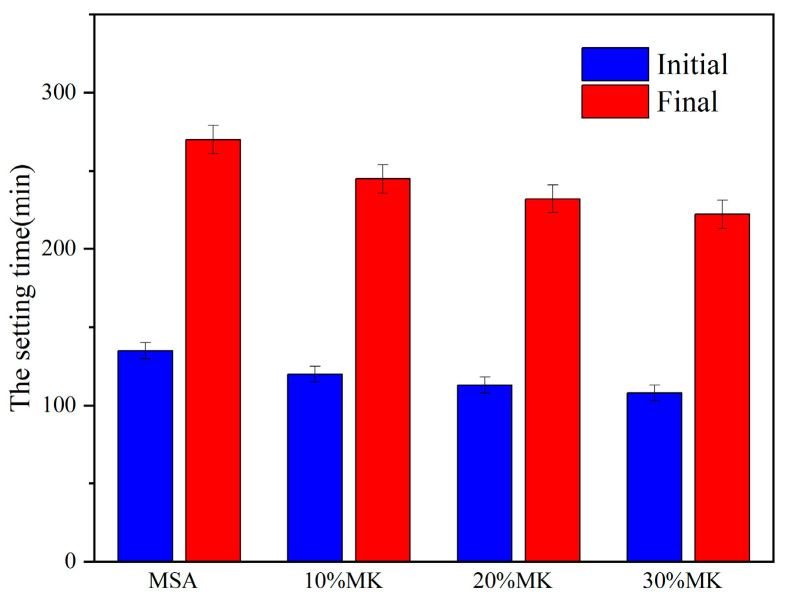
Effect of MK on the setting time of MSA cement slurry.

**Figure 7 materials-17-04567-f007:**
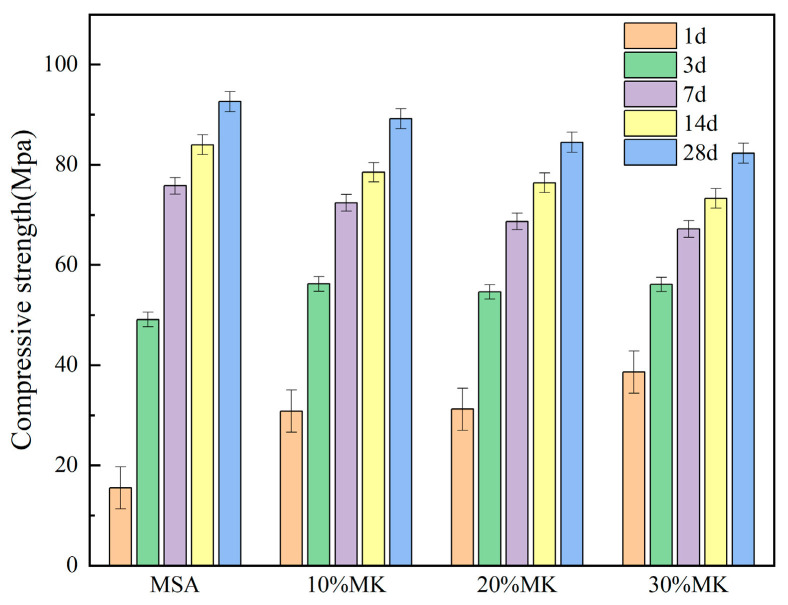
Effect of MK on the compressive strength of the MSA cement.

**Figure 8 materials-17-04567-f008:**
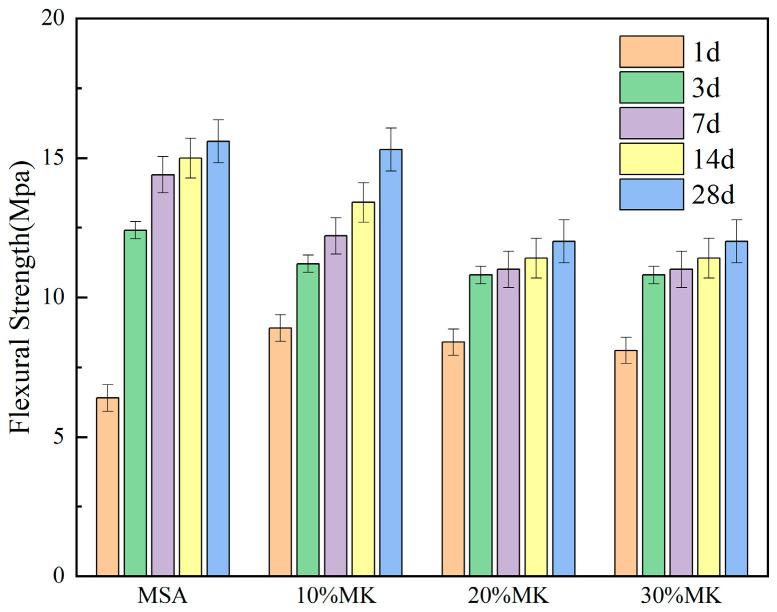
Effect of MK on the flexural strength of the MSA cement.

**Figure 9 materials-17-04567-f009:**
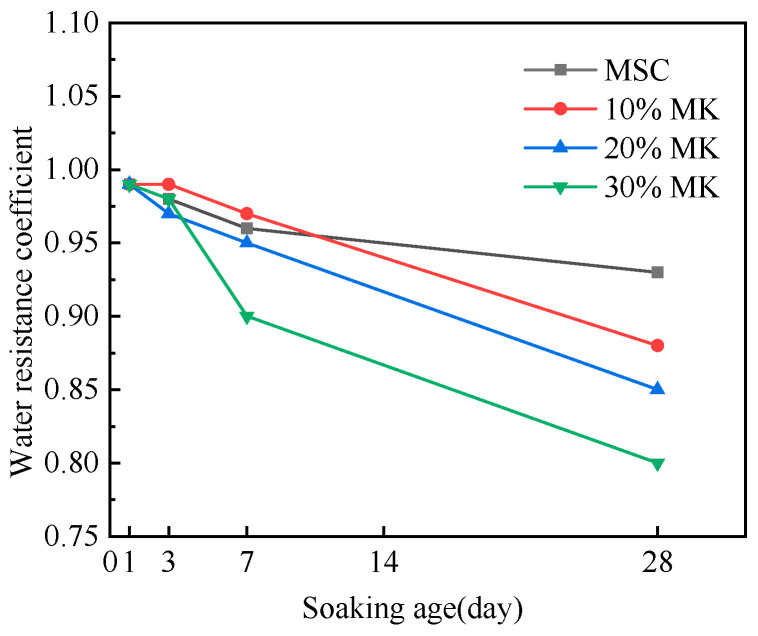
Effect of MK on the water resistance of the MSA cement.

**Figure 10 materials-17-04567-f010:**
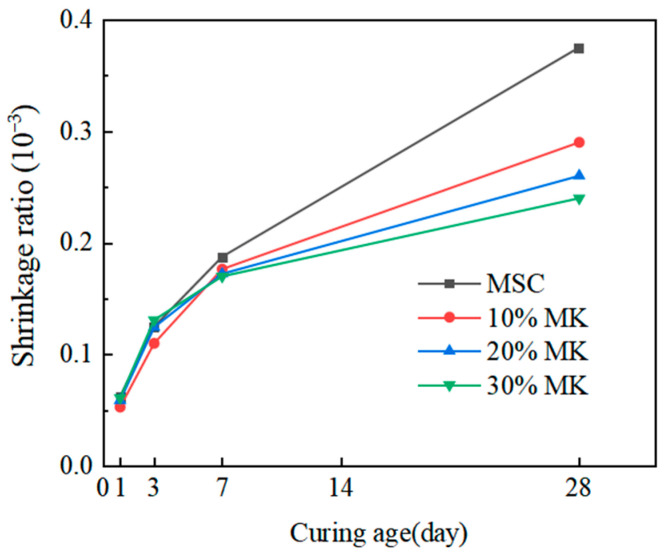
Effect of MK on the shrinkage of the MSA cement.

**Figure 11 materials-17-04567-f011:**
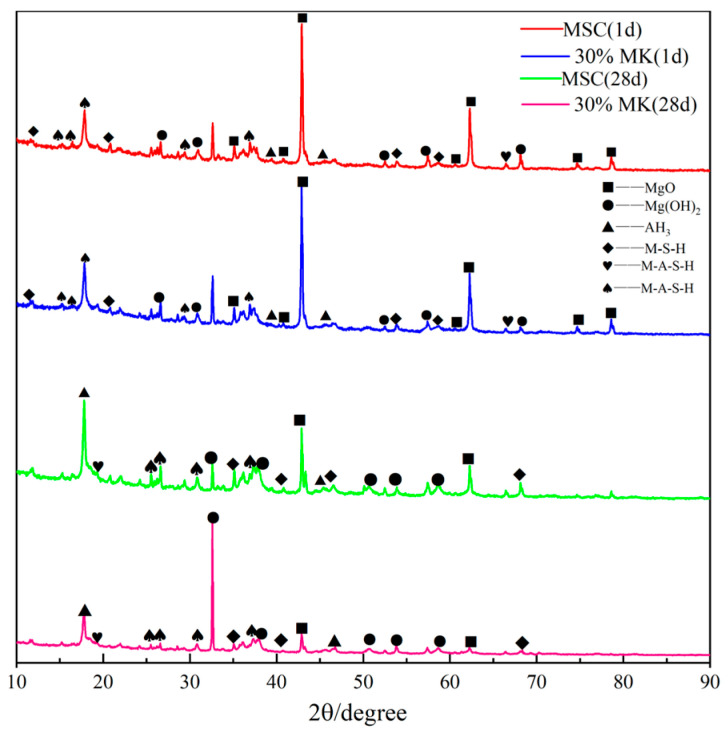
XRD patterns of the MSA cement at the curing ages of 1 d and 28 d.

**Figure 12 materials-17-04567-f012:**
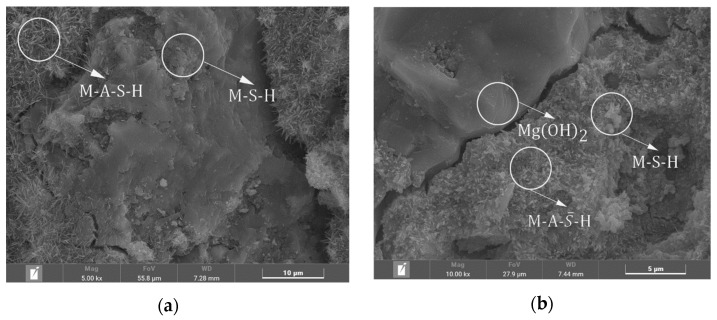
SEM images of the MSA cement at the curing ages of 1 d and 28 d. (**a**) 30%MK (1 d); (**b**) MSA cement (1 d); (**c**) 30%MK (28 d); (**d**) MSA cement (28 d).

**Figure 13 materials-17-04567-f013:**
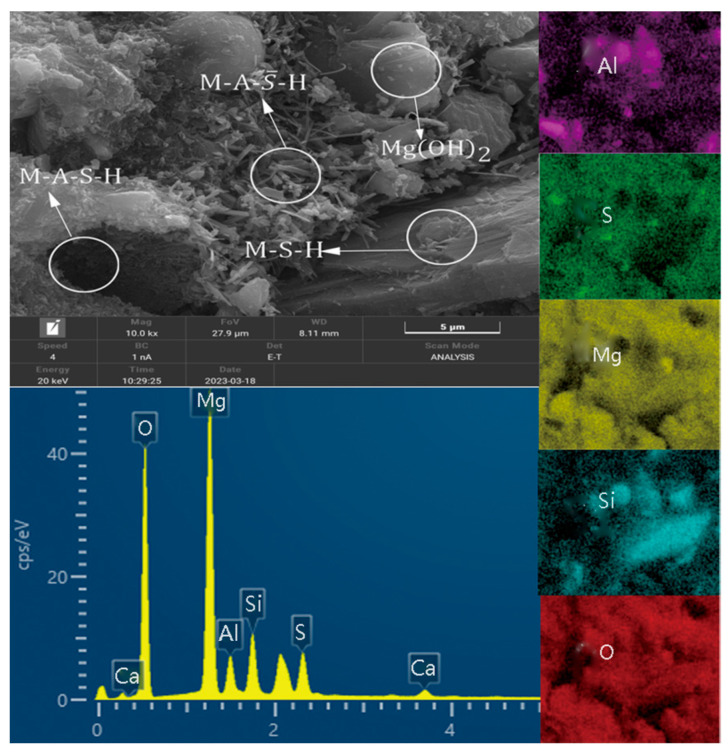
EDS image and elements of the MSA cement with 30%MK at a curing age of 28 d.

**Table 1 materials-17-04567-t001:** Chemical compositions of lightly burned magnesium oxide.

Component	MgO	SiO_2_	CaO	Al_2_O_3_	Fe_2_O_3_	Ignition Loss	Other
Content (%)	88.12	3.6	1.3	0.77	0.68	1.6	3.93

**Table 2 materials-17-04567-t002:** Chemical compositions of bauxite and metakaolin.

Component		SiO_2_	Al_2_O_3_	CaO	Na_2_O	Fe_2_O_3_	MgO	SO_3_	TiO_2_	K_2_O
Content (%)	Bauxite	39.48	53.33	0.92	0.13	1.75	0.35	0.89	1.8	0.41
	MK	49.94	43.88	0.27	0.00	0.51	2.66	0.14	1.89	0.23

**Table 3 materials-17-04567-t003:** Element distribution map.

Elements	Apparent Concentration	K Ratio	wt.%	wt.% Sigma	At%
O	53.58	0.180	59.79	0.11	70.60
Mg	19.34	0.128	26.11	0.08	20.29
Al	2.29	0.016	4.01	0.04	2.81
Si	3.61	0.028	5.38	0.04	3.62
S	2.96	0.025	3.94	0.04	2.32
Ca	0.75	0.006	0.77	0.02	0.36

## Data Availability

The original contributions presented in the study are included in the article, further inquiries can be directed to the corresponding author.

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
