# Peer review of "The Effects of Metakaolin on the Properties of Magnesium Sulphoaluminate Cement"

_materials, 2024, doi:10.3390/ma17184567_

Round 1
Reviewer 1 Report
Comments and Suggestions for Authors
Manuscript ID: materials-3195581-R1. Title: Effect of Metakaolin on Properties of Magnesium Sulphoaluminate Cement.
Comments:
1. The similarity percentage of this article is high (iThenticate): 24%. The authors should reduce this similarity index below 20%. Otherwise, I suggest rejecting the article for publication.
2. Please visualize in the abstract the practical utility of this study.
3. The introduction of the article should be significantly improved. Please include the practical usefulness of this study.
4. In the introduction please include information regarding the following technical aspects: Chemical composition and microstructure (detailed chemical analysis and microstructural characterization), mechanical properties (compressive strength, modulus of elasticity and toughness), hydraulic properties (heat of hydration, rate of hydration and permeability), durability (sulfate resistance, chloride resistance and resistance to freeze-thaw cycles), rheology, shrinkage and expansion.
5. Please check all the wording and style of the article. There are errors. For example: "Magnesium Oxude".
6. This article contains a large number of abbreviations. These should always be explained the first time. In addition, authors should include a final section with all abbreviations used in this article.
7. In Figure 2 please include a legend to indicate what each line color corresponds to.
8. Please integrate figures 3 and 4.
9. Chapter 2, which should be titled Materials and Methods, has a large number of very short sections. Please integrate and reduce the number of sections in this chapter.
10. Authors should include a figure with a representative photograph of the experimental design used in this study.
11. In Chapter 2, the authors should include a section with all the statistical analyses used in this study. In addition, did the authors use any statistical software for this?
12. Figures 11 and 12 could be integrated. Currently, the article has a large number of figures. Please evaluate the integration of some of them or use the supplementary material option appropriately.
13. Figures 13 and 14 can also be integrated.
14. The chapter on results and discussion should be significantly improved. Although the authors adequately presented all the required analyses (chemical composition, microstructure, etc.), there is no discussion of their results in the light of other reference authors. It is necessary to contrast the results of their study with other authors in order to have a more in-depth discussion of the results.
15. Please quantify the findings presented in conclusion number one.
16. Please close the conclusions with the main limitations detected during the development of your study. Additionally, future lines of research should be included.
17. References used in this article should be more recent. Please improve this aspect in your article.
18. In general terms, this article should be significantly improved. In the introduction there is relevant information to be included, the chapter on materials and methods should have more technical detail, and the chapter on results and discussion should have more scientific depth. It is important that the authors do not lose sight of the following technical factors throughout these chapters: Chemical composition and microstructure (detailed chemical analysis and microstructural characterization), mechanical properties (compressive strength, modulus of elasticity and toughness), hydraulic properties (heat of hydration, rate of hydration and permeability), durability (sulfate resistance, chloride resistance and resistance to freeze-thaw cycles), rheology, shrinkage and expansion.
Comments on the Quality of English Language5. Please check all the wording and style of the article. There are errors. For example: "Magnesium Oxude".
Reviewer 2 Report
Comments and Suggestions for Authors
The article relates to the influence of metakaolin on physical and mechanical properties of magnesium sulphoaluminate cement. The properties were analyzed by X-ray diffraction, scanning electron microscope, energy dispersive X-ray spectroscopy, Fourier transform infrared spectroscopy and thermal analysis-thermogravimetric. The achieved results show an important impact of the adopted parameters.
My comments are as follows:
1.What important goal was achieved in the research that the authors published the observed dependencies in cement properties? Add this statement to Abstract.
2.Figure 4 is missing, it is not mentioned in the text either. Probably a mistake in numbering.
3.Reference to a larger number of contemporary literature items (from the 2020s) seems necessary.
4.What does the term: "Intensity(a.u.)" on the x-axis mean - Figure 1? In what units is it measured?
5.What do: "Cum(%) and Diff(%)" mean in Figure 2?
6.Improve English language - Line 110; ... pastes were use ....
7.A block chart illustrating the research methodology, including the workflow and types of parameters - dependent and independent variables, would be useful.
Comments on the Quality of English LanguageEnglish Language should be improved, e.g., Line 110; ... pastes were use ....
Reviewer 3 Report
Comments and Suggestions for Authors
Title: Effect of Metakaolin on Properties of Magnesium Sulphoaluminate Cement
Decision: Major revisions.
General comments:
- The abstract needs to be revised, including the main experimental results obtained in the research and including the main conclusions of the research.
- “Magnesium oxysulfate (MOS) cement has the advantages of excellent performance, 27 energy saving, emission reduction and so on[1-3]”. So on? It seems that the information is incomplete, please review.
- The article presents many generic citations, where the authors use references to discuss information widely known by authors in the field (first paragraph, for example). Please review the use of generic citations in your article.
- “In 2013, researchers including Professor Hongfa Yu and Dr Chengyou Wu[6,7].” These citations are incorrect, this is not how citations are used in research articles. Please revise accordingly.
- “According to the experimental exploration and production experience, Meng Chen et al. [15,16] added volcanic ash material into the MgO-MgSO4-H2O cementitious system.” I don’t understand why the authors always include two citations for each piece of information. Are the two studies similar? Why cite two references for the same information? Please clarify.
- “In this paper, metakaolin was used to partially replace bauxite powder in MSA cement.” Please clarify the innovations in your research. What are the knowledge gaps being addressed in your paper? Please make this clear in the text of the manuscript. Explain why your paper is an original contribution. For example, what are the differences between your research and Effect of metakaolin and magnesium oxide on flexural strength of ultra-high performance concrete? Please clarify this in the text of the manuscript.
- The title of the manuscript and the entire introduction highlight the use of MK in MSC, but the materials section contains information about the use of bauxite in the research. This is so bad because at no other time was it clear that this material would be evaluated in your work. Please correct this information and make it clear which materials were used in your research throughout the text.
- The MK used has many crystalline phases, which were named by the authors as Al2O3 and SiO2. Please correct this by specifying the corresponding mineral, probably kaolinite and quartz. Also make it clear in the text of the manuscript that for MK to perform adequately, the material must be predominantly amorphous, see for example in Recycling of calcined clay as an alternative precursor in geopolymers: A study of durability. Include a comment on the limitations of the research due to this.
- Table 3 is not clear, use the specific quantities of materials used in the research, including each component and specifying how the dosage was performed in your research. Also explain why the authors replaced bauxite with MK and why the replacement percentages of 10, 20 and 30% were chosen. Please improve the information on dosage performed in Table 3.
- The discussion of the results is very poor, the authors do not use bibliographical references in the discussion of results section, which is totally unacceptable for an experimental article. This indicates that the results obtained were not compared to similar research and that the theoretical information described in the article is not based on other scientific texts. This is very bad. In order for the article to be considered for publication, it is necessary to improve the discussion of results section, otherwise the article is not useful.
- Include the experimental deviation bar in all results obtained in your research. Please correct this appropriately.
- Figure 5: without the deviation bar, it is difficult to understand and discuss the patterns obtained in the research. Please include the deviation bar and take this into account in the discussion of your results.
- “Metakaolin has undergone high-temperature calcination and the particles display a disordered structure and have larger specific surface area. The water requirement of the cement increase with the addition of metakaolin. Thus, the irregular particle morphology of metakaolin increases the inter-particle friction and significantly reduces the fluidity and shortens setting times of the cement.” This information is not related to the experimental results of your research, so it is necessary for the authors to include a reference to this information.
- Figure 7 and 8: in general, it is observed that the effect of MK is to accelerate the hydration of the cement, although it is expected that pozzolanic materials have a more delayed hydration and gain in strength in the case of OPC. In the case of MAS this seems to be different, and may be related to the morphology of MK, discussed, for example, in Recycling of calcined clay as an alternative precursor in geopolymers: A study of durability. Please explain this effect better in the manuscript text.
- “Metakaolin belongs to amorphous layered aluminosilicate mixture with high hydration activity. This activity accelerates hydration rate and creates a number of gel substances quickly, including magnesium aluminosilicate hydrate (M-A-S-H) due to the action of OH- and SO4 2- . This process significantly improves the compactness of the system. Thus, the initial strength shows a noteworthy enhancement in comparison to the control sample.” This information is not related to the experimental results of your research, so it is necessary for the authors to include a reference to this information.
- Review the results of Figure 12. Apparently some peaks related to the presence of quartz were attributed to other phases. Review this information accordingly.
- Results in Figures 16 and 17 are unnecessary and do not add useful information to your research. The objective of your work is to evaluate the effect of MK on MSA cement. What do the results of these figures add to the discussion about this objective? If the authors include information about the compositions 10, 20 or 30% the results are useful. Otherwise they are unnecessary because they are not related to the main objective of your study. Please review and delete this information.
- Review the conclusions after reviewing the previous comments. If necessary, change the conclusions appropriately.
Round 2
Reviewer 1 Report
Comments and Suggestions for Authors
Manuscript ID: materials-3195581-R2. Title: Effect of Metakaolin on Properties of Magnesium Sulphoaluminate Cement.
Comments:
1. In this new version of the article, the authors adequately addressed each of the 18 comments made in the previous review. Therefore, I suggest acceptance of the article, after a new similarity check using iThenticate.
Reviewer 3 Report
Comments and Suggestions for Authors
Accept in present form